# Beyond partisan filters: Can underreported news reduce issue polarization?

**Curtis Bram** [ID] *

The University of Texas at Dallas, Richardson, TX, United States of America

* curtis.bram@utdallas.edu

## Abstract

While many news outlets aim for impartiality, 67% of Americans perceive their news sources as partisan, often presenting only one side of the story. This paper tests whether exposing individuals to news stories their political adversaries focus on can mitigate political polarization. In an experiment involving a real-world political newsletter—sent to participants who had opted to receive news that uncovers media biases—exposure to a specific story about refugee policy led respondents to reassess their positions. This reevaluation changed their stances on the issue and reduced the ideological distinctions they made between Democrats and Republicans. These findings underscore the need for future studies to untangle the specific circumstances where cross-partisan exposure can alter political attitudes.

## Introduction

The motto of Fox News reads "Fair and Balanced", and that of The New York Times is "All the News That's Fit to Print." Journalists and news sources often claim to be neutral. However, the public is largely skeptical of these claims—a 2020 Pew Research poll finds that a striking 67% of Americans think their preferred news sources present facts with a partisan slant. These survey responses reveal a widespread awareness of partisan biases in news coverage. In addition to this awareness, researchers find that partisan media can contribute to polarization [1–4]. If people suspect that their preferred sources are biased, then can revealing coverage biases reduce issue polarization?

When a Republican only watches Fox News, or a Democrat only watches MSNBC, that person receives their news from like-minded sources (AllSides rated those sources as right- and left-leaning, respectively, in 2023). This person is then in a news "echo chamber", which is where people primarily or exclusively consume content from politically aligned sources. The potential problem is obvious; if right-leaning sources highlight all of the mistakes the left is making, and left-leaning sources highlight all of the mistakes the right is making, then both Republicans and Democrats will have completely divergent (and possibly wrong) views of each other. In fact, recent research shows that focusing only on the issues that politicians disagree on causes people to misperceive the viewpoints of Democrats and Republicans, promoting polarization [5]. This finding suggests that prompting news consumers to broaden their focus—meaning to break out of their news echo chambers—can reduce polarization.

**Data Availability Statement:** Replication data and code are available at: https://doi.org/10.7910/DVN/QLF6MV.

**Funding:** Duke University provided funding for the incentive for participants to complete the surveys, and I later received a grant from the Institute for

Humane Studies to fund the publication of this paper (grant: # IHS017528). The funders had no role in study design, data collection and analysis, decision to publish, or preparation of the manuscript.

**Competing interests:** I worked with Ground News to implement this experiment. To minimize the appearance of a conflict of interest, I never received compensation from the company. When I presented this work together with Ground News employees at the Online News Association's 2022 conference, funding for travel was provided by Duke University; the conference registration fee was covered by Ground News. Finally, Ground News provided data on whether study participants opened the newsletter, but never had access to individual survey responses. Ground News was not involved in the analysis of these data, but has had the chance to review this manuscript to ensure that their products and business are accurately described. The company did not ask for any changes after reviewing the paper.

There has been research on echo chambers on social networks [6]. Experimental tests of potential echo chamber effects often use simulated information environments, so it is difficult to know whether these findings generalize to a real-world environment [7, p. 138]. To rectify that issue, a component of the 2020 Election Research Project reduced real-world exposure to like-minded sources on Facebook and found that doing so did not affect polarization or other attitudes [7]. That line of research has focused on social media, probably because social networks give consumers tools to easily select their own information environments.

In addition to more recent work focused on social networks, studies dating back to the 1960s emphasize the importance of selective exposure to political information, when individuals opt to consume news that reinforces their preexisting beliefs [8]. This selective exposure goes beyond social media, contributes to polarization [9, 10], and intensifies before elections [11, 12]. People often select information that fits their worldview, thereby sorting into polarized groups [13, 14].

Because the tendency to consume information from like-minded sources extends beyond social media, it is important to explore how exposing news consumers to information from stories across the partisan divide affects polarization. In this paper, I unobtrusively randomized the content of a real-world political newsletter. The experiment was carried out with a highly selected group of participants who had registered to receive information about stories ignored by left and right-leaning sources. By manipulating the content of these newsletters, I aim to assess the impact of exposure to specific stories on both political issue polarization and individual issue stances.

The results indicate that one of the four stories tested encouraged a shift in perspective. This suggests that, at least in the context of short-term exposure, issue polarization is not solely attributable to a lack of awareness about the topics covered by opposing political sides. These results align with studies of online partisan media that report that exposure to opposing media produces minimal effects [15] or that the effects of partisan media may be overstated [16]. Importantly, recent studies have found mixed evidence on the existence of online echo chambers [17]. Furthermore, politics is not a central part of most people's lives, and the debate continues over the societal effects of changes in the digital media landscape [18, p.,16]. However, these findings are also consistent with recent research demonstrating that transitioning partisan media consumers to a source with opposing bias can lead to significant shifts in attitudes about some issues [1, p. 9].

Given the overall results, a key takeaway from this work is the variation in the circumstances under which examining partisan selective exposure can impact attitudes. Since at least one story may have caused people to change their minds, future research should explore the conditions that determine the extent and manner in which polarized media influences political discourse and outcomes. By identifying and understanding these nuances, we may be better equipped to address the challenges posed by partisan media in contemporary society.

## Experimental design

To examine how information about media coverage biases affects people's attitudes toward political issues, I used real-world treatments, delivered through an emailed newsletter to which participants had already subscribed. These treatments were incorporated as part of Ground News's weekly "Blindspot Report," showcasing stories from the preceding week that received minimal attention from one political side or the other. Stories overlooked by one political side are included in the newsletter, which was distributed to approximately 115,000 subscribers on Tuesday evenings at the time the experiment was conducted. All Blindspot Report readers had either self-subscribed or provided their email when signing up to use Ground News's mobile

application. This means that all participants in this study were Ground News subscribers. They had willingly elected to receive a newsletter that focuses on illuminating overlooked stories from various partisan viewpoints.

Although there is currently no way to definitively compare Ground News's subscribers to the population, the company's approach to news coverage suggests that its subscribers possess an intrinsic interest in exploring narratives beyond conventional partisan lines. Consequently, they could be more receptive to persuasion and might exhibit greater flexibility in their attitudes and beliefs than the general populace. Furthermore, their subscription to a service that reveals overlooked stories indicates a level of engagement with broader political news that may not be as prevalent among the general public. This distinguishing characteristic of the sample —their openness to cross-cutting information—may influence the study results and shape how these respondents react to different types of stories they receive. This sample may be a particularly engaging group for exploring questions of persuasion and attitude change in the political sphere. At the same time, the uniqueness of the sample means that it is not possible to generalize the results of this experiment to the population.

At the time the experiment was run, Ground News processed more than 50,000 news articles each day, utilizing proprietary natural language processing algorithms to cluster articles from numerous news outlets into unified news stories to reveal coverage biases. The company rates bias at the source level (for instance, Fox News is classified as right-leaning and the New York Times as left-leaning). To develop these bias ratings, Ground News averages ratings from three independent, nonpartisan media monitoring organizations (AllSides, AdFontes Media, and Media Bias Fact Check). The company takes that average rating and maps it onto a seven-point ideological spectrum, with sources ranging from strongly left-leaning to strongly right-leaning.

To run the experiment, I selected four breaking stories from July 14th, 2021, to July 18th, 2021, that received limited coverage from either conservative or liberal-leaning media. These issues received most of their attention from partisan sources. To select these stories, I deviated from Ground News's usual approach to story selection. Since the stories had to be chosen just a day before the first part of the survey was administered, I based the selection on the political importance of the issues at the time the experiment was conducted. Selecting stories in this way may have affected the experimental results. This approach to story selection could be systematized in future work, but was necessary due to the constraints imposed by utilizing the real-world newsletter in the experiment. For example, as demonstrated in Fig 1, while CNN covered the story about the Biden administration's evaluation of the origins of Covid-19, most of the sources who chose to discuss the story leaned right. The explanation for this could be that left-leaning outlets saw the story as a policy reversal potentially harmful to President Biden, which may have led some partisan sources on the left to choose to ignore the story.

The stories overlooked by the left included the Biden administration's decision to return refugees fleeing Cuba by sea (9% left-leaning coverage) and the Biden administration's assessment that the virus causing the Covid-19 pandemic is equally likely to have originated from a lab leak as from direct contact with animals (13% liberal coverage). Right-leaning sources neglected Republican Governor Spencer Cox's statement that vaccine misinformation is lethal (10% conservative coverage) and the leak of Kremlin papers revealing Putin's plot to install Trump in the White House (25% coverage from the right). Fig 1 shows an example of a story from a randomly assigned newsletter that was predominantly ignored by left-leaning media. Due to the design of Ground News's newsletter, treatment stories encapsulate information about the story itself and the partisan breakdown of coverage.

On Monday, July 19th, subscribers to the Blindspot Report were invited to take a baseline survey. This survey encompassed questions about all four issues; their ideology, their country

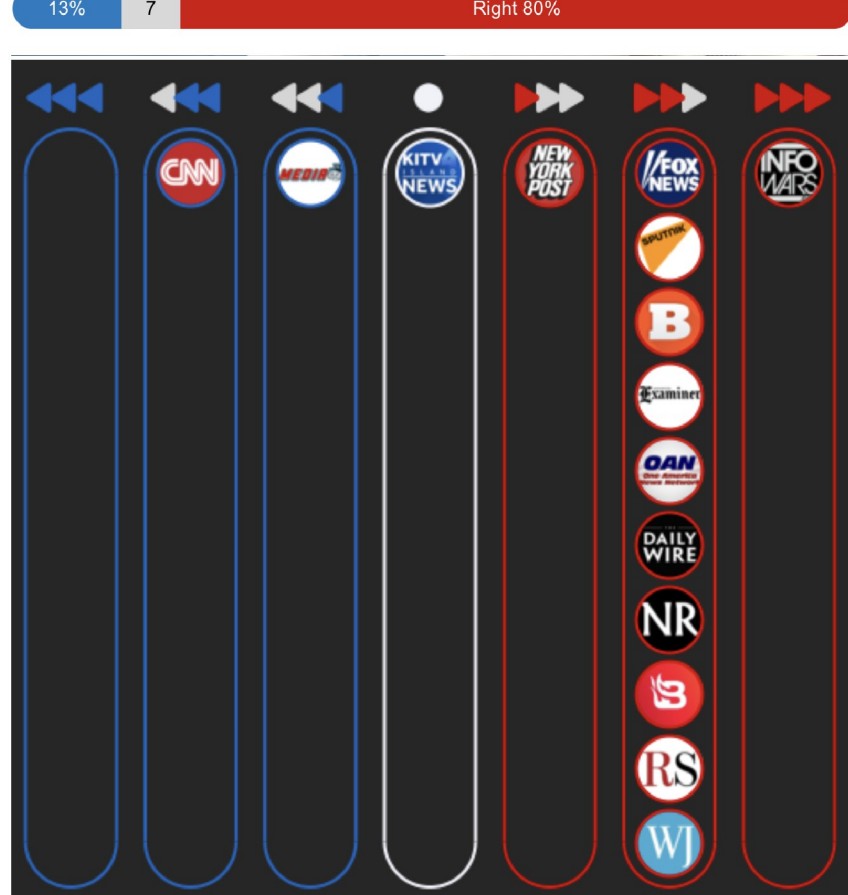

**Fig 1. Example of a story that received differential coverage from liberal- and conservative-leaning sources, and which was also included as an experimental treatment.** Reprinted from the original under a CC BY license, with permission from Ground News, original copyright 2023. Please note that the original figure included a photo that was omitted in this paper for copyright reasons.

of origin, and their partisanship if they hailed from the United States. All respondents read and agreed to an informed consent form before starting each survey. This study received approval from Duke University's Institutional Review Board on July 8, 2021 (protocol number: 2021–0598) and did not involve any deception. Of the initial 2,372 respondents, 1,861 reported an email address corresponding to a Blindspot Report subscriber. This discrepancy likely arises

because respondents were asked to self-report their email address for the purpose of contact in the recontact study, leading to some entering a different email address from that used to subscribe to Ground News's content.

This experimental design aligns with previous work advocating that field experiments employ repeated measures designs with samples drawn from known populations [19]. Questions about the four issues remained consistent throughout both survey waves and were structured identically across the issues. The rationale behind this pre-post approach was to enhance precision, and previous work validates that these designs achieve this without biasing respondents [20]. There is little evidence that respondents discern the purpose of experimental studies and alter their behavior accordingly (commonly called demand effects) in political science and economics experiments [21, 22].

Both iterations of the newsletter were sent to all participants on July 20th, 2021. The following day, all respondents from the initial wave were invited by email to participate in a follow-up survey that was substantially identical to the first wave but incorporated additional demographic questions, such as age, sex, race, and education. The final sample comprised the 1,234 individuals who received treatment newsletters and completed both surveys. The average respondent was 44 years old and had a Bachelor's degree. Among the respondents, 82% identified as white and 74% were men. Of the 921 US-based respondents, 222 (24%) identified themselves as Republicans, 383 (42%) as independents, and 316 (34%) as Democrats. Respondents were evenly distributed across conditions.

## Results

This research examined four significant news stories: Cuban refugees arriving by sea, vaccine misinformation, the origins of the Covid-19 pandemic, and Russian election interference. All dependent variables were measured before and after exposure to the news stories, and the change in responses serves as the dependent variable for all analyses. I focus on four attitudes toward each issue and these represent all the measured outcomes in the experiment. These are the personal importance individuals place on the issue, their stance on the issue, their beliefs about the extent of polarization among politicians on the issue, and their approval of Democratic and Republican politicians' handling of the issue. I coded changes in issue positions so that a positive change means that the respondent moves his or her position toward the perspective implied by the story. For example, a positive change in position toward vaccine information means that the respondent wants the government to exert more effort to combat vaccine misinformation. Of course, describing these changes as positive or negative in this study does not imply anything about the normative benefits or costs of these positions.

The Online S1 Appendix (Section 6.1) reports the experimental results for each possible attitude-issue combination. I find little evidence of uniform change in response to these four stories, with no p-value less than a Bonferroni corrected significance threshold of 0.003125—the correction is used to compensate for the fact that there are 16 total attitude-issue combinations. But these average effects cannot reveal much about how people truly responded to the stories. The reason is that the Blindspot report highlights stories that partisan media consumers would otherwise miss. Therefore, what matters is whether the stories fall into someone's Blindspot and are thus likely to challenge people's perspectives. For example, Republicans and Democrats may change their attitudes in completely different ways when they see a story indicating that President Biden is less welcoming toward Cuban refugees than they likely expected.

To learn about how people respond when they receive a story that challenges their own ideological priors, I pool the issues together and focus on changes in the four measured attitudes, aggregating the four issues so that there are then four total tests. I explore the influence

of the type of story received on the change in response for each question type. This analysis takes into account the clustering within respondents using robust standard errors that account for potential heteroskedasticity.

I categorize respondents based on the type of story they received. First, the baseline category (not receiving a story) corresponds to the over-time change in opinion, because I have pre-post data on respondents' attitudes for both the stories that they did and did not receive. This categorization is important because attitudes towards each issue may have changed over the days the study was conducted, perhaps because of incidental exposure to stories from other sources. Next, American partisan respondents can receive a "Blindspot" story, such as when Democrats read about Biden's policy toward Cuban refugees or the origins of the Covid-19 pandemic. On the other side, a Blindspot for Republicans is when they read about vaccine mis-information or Russia's election interference. Partisan respondents can also receive an "in-partisan" story. Those stories are the reverse of the Blindspot story—Democrats read about vaccine misinformation and Russia while Republicans read about Cuban refugees and the origins of Covid-19. In-partisan stories can be considered analogous to traditional partisan media. When people receive these stories, they are often reinforcing their original beliefs. Finally, American independents and those located outside the U.S. can receive stories.

Table 1 shows the results of this analysis considering the type of people and the type of stories they receive. Again, the data are aggregated into long-form to reflect the fact that everyone received four stories and answered questions about all four attitudes. All respondents fit into one of the included categories, reflecting their receipt of a story and their own position relative to that story. In examining the table, several findings deserve particular attention. When no story is presented, which serves as the change-over-time baseline, there is a small but statistically significant increase in issue polarization ($0.08$, $p < 0.05$) and approval of politicians ($0.01$, $p < 0.05$). For Democrats who received a Blindspot story, issue polarization appears to

**Table 1. Shows how respondents in each group respond to the aggregated issues.**

|  | Issue polarization | Positions | Importance | Approval |
|---|---|---|---|---|
| Baseline (no story) | 0.08* | −0.00 | −0.00 | 0.01* |
|  | [0.07;0.09] | [−0.01;0.01] | [−0.01;0.00] | [0.00;0.02] |
| Democrat received Blindspot | −0.07* | −0.00 | 0.01 | −0.00 |
|  | [−0.10; −0.03] | [−0.03;0.02] | [−0.01;0.03] | [−0.04;0.03] |
| Democrat received in-partisan | 0.01 | 0.01 | 0.00 | −0.02 |
|  | [−0.02;0.04] | [−0.01;0.03] | [−0.01;0.01] | [−0.05;0.02] |
| Independent received story | −0.00 | 0.02* | 0.01 | −0.02 |
|  | [−0.02;0.02] | [0.00;0.04] | [−0.01;0.02] | [−0.04;0.01] |
| Non-U.S. respondent received | −0.00 | 0.00 | −0.00 | 0.01 |
|  | [−0.03;0.02] | [−0.02;0.02] | [−0.02;0.01] | [−0.01;0.04] |
| Republican received Blindspot | 0.04 | 0.02 | 0.01 | −0.02 |
|  | [−0.00;0.09] | [−0.01;0.05] | [−0.02;0.04] | [−0.06;0.03] |
| Republican received in-partisan | −0.01 | −0.03* | 0.02 | −0.01 |
|  | [−0.05;0.03] | [−0.06; −0.00] | [−0.00;0.05] | [−0.06;0.03] |
| $R^2$ | 0.00 | 0.00 | 0.00 | 0.00 |
| Adj. $R^2$ | 0.00 | 0.00 | −0.00 | −0.00 |
| Num. obs. | 4921 | 4934 | 4935 | 4921 |
| RMSE | 0.28 | 0.22 | 0.17 | 0.29 |
| N Clusters | 1233 | 1234 | 1234 | 1233 |

*Null hypothesis value outside the 95% confidence interval.

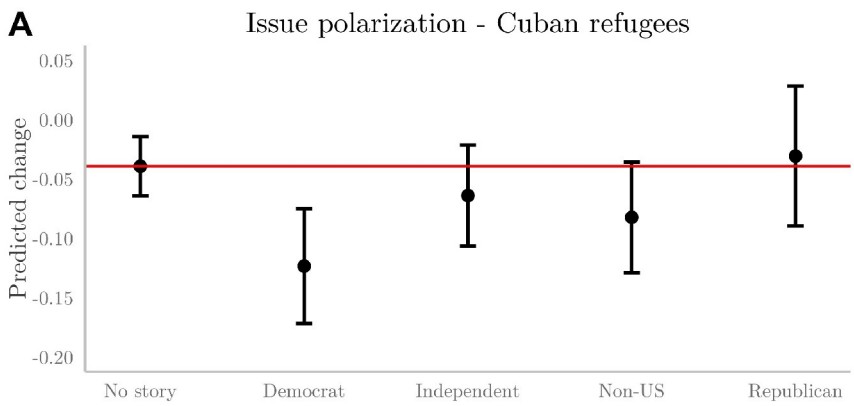

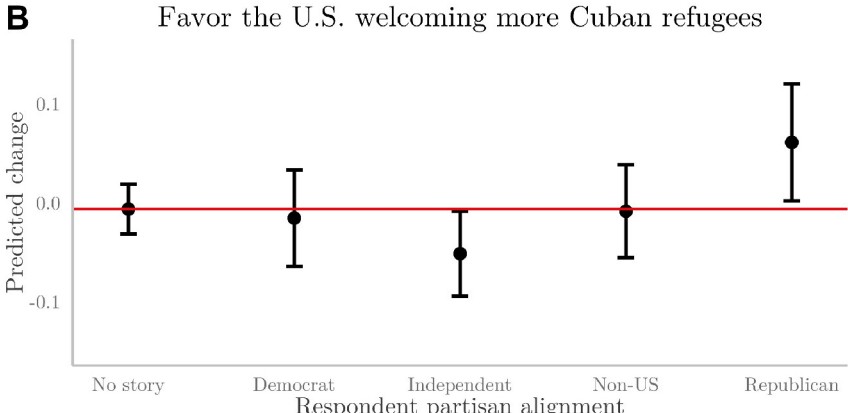

**Fig 2. Shows the change in people's reported issue polarization (A) and issue positions (B) for the Cuban refugee issue.** Categories are a baseline for the control group that did not receive a story (representing the change in people's responses to outcome questions across waves). Then, indicators for each possible group are included. Error bars are 95% confidence intervals, and the red line indicates the mean of the baseline category; deviations from this mean signify statistically significant results.

decrease ($-0.07$, $p < 0.05$), and there are no appreciable shifts in other attitudes. Independents slightly adjust their issue positions ($0.02$, $p < 0.05$) when exposed to a story, suggesting some receptivity to change. Meanwhile, Republicans who encountered in-partisan stories also slightly but significantly adjusted their positions ($-0.03$, $p < 0.05$). The metrics on importance and approval mostly remain stable across different treatments. Across all models, the share of variance explained by the experimental treatments is very low. Overall, these data indicate that the type of story can exert targeted, although modest, impacts on select political attributes.

These aggregated results indicate that Democrats may feel like there is less issue polarization when they receive a Blindspot story and that Republicans may adjust their position in response to partisan stories. Because those results are aggregated, they combine all stories. To understand exactly what stories drive these changes, it is important to disaggregate the analysis and examine each story separately. When looking at specific stories, I find that three of the four stories did not change people's minds on these issues. However, the story about the Biden administration's policy toward Cuban refugees significantly reduced issue polarization among Democrats and significantly increased Republicans' favorability toward welcoming these refugees (Fig 2). These findings are substantively consistent when restricting the data for analysis to the 642 respondents for whom Ground News confirmed had opened the newsletter when the recontact survey request was sent: issue polarization among Democrats receiving Cuban

story: ($\beta = -0.16$, S.E. = 0.04, $P < 0.001$); attitudes toward welcoming more Cuban refugees among Republicans receiving Cuban story: ($\beta = 0.08$, S.E. = 0.05, $P < 0.07$).

Finally, beyond looking at specific stories, one additional benefit of this design was that 308 people completed the initial survey and follow-up, but had provided an email address that did not match an existing Blindspot Report subscriber. These individuals received a newsletter that did not feature any of the stories included in the experiment. Those respondents had been assigned to receive one of the two randomized newsletters, but because their emails did not match that of a current subscriber, they were sent a newsletter with no issues related to the survey. As expected, the random assignment among the partisans in this group, which did not receive treatment, did not significantly affect attitudes.

## Discussion

Despite the seemingly high political stakes surrounding Russian election interference, the lab-leak hypothesis, and vaccine misinformation, information on these issues did not appear to sway opinions in this study. However, both Democrats and Republicans reacted to a story about refugee policy. This story differs from the other three in that it describes an unexpected outcome. During the 2020 campaign, the then-candidate Biden had pledged to allow more refugees to settle in the United States [23]. The administration's apparent decision to impose greater restrictions on these refugees was probably surprising to partisans from both sides. As a result, Democrats seem to report lower levels of observed issue polarization, perceiving their party's position as less distinct from Republicans'. Conversely, Republicans seem to respond by favoring increased openness to these refugees.

Future work will expand the scope of this research beyond a single week and beyond the four issues studied in this experiment. The manipulation studied in this trial was only done once, and one can imagine different effects were people to consistently receive information about their Blindspots on one or many issues over time. However, even such an experiment would be limited by the fact that most people spend little of their time online engaging with political news; recent work finds that news exposure takes only about 3% of people's time online [24]. That said, one key aspect of this study's design lies in its execution among individuals who one would expect to be most receptive to altering their perspectives. All participants in this research had demonstrated their willingness to engage with potentially challenging information outside of the experimental context. Given that each participant had voluntarily opted to receive stories that might cast doubt on their own partisan side, one might anticipate that this group would be especially prone to changing their minds.

A second notable limitation of this experiment is that the results focus exclusively on policy issues. Scholars investigating contemporary politics debate whether polarization has its roots in ideological or identity-based foundations [25, p. 59]. In accordance with this, recent research reveals that warm personal relations between political leaders can reduce partisan hostility, while policy compromise does not have the same effect [26], suggesting that animosity is not primarily driven by policy differences. Similarly, when it comes to online echo chambers, the presence of segregated online groups correlates with negative online interactions [27]. Isolated online groups may be more prevalent among conservatives [28], and the right can have an advantage in social media sharing [29]. But at the same time, banning content can worsen social media platforms [30] and having to reverse course after relegating certain views to the fringes [31] can affect public opinion.

While the Blindspot Report focuses primarily on substantive political stories, it might be assumed that the newsletter would only influence attitudes on policies, mirroring this paper's focus on issues. However, future research could explore the effects of correcting media

Blindspots on attitudes that extend beyond the issues. A grid question asking participants to evaluate the Blindspot Report was included in the recontact wave after all other questions had been answered. On average, respondents felt that the newsletter helped them become aware of others' biases, understand more about views they disagree with, recognize their own biases, and become more empathetic towards political opponents. These responses encourage further investigation into how policy information can mitigate hostility between partisans.

## Conclusion

In conclusion, these results indicate variation in how people respond to different types of stories, a finding that complements recent work attempting to understand how partisan media can influence attitudes [1, 2]. The observed heterogeneity in reactions introduces another layer to the ongoing debate on the role of partisan media in shaping public opinion [3, 4]. The fact that respondents seem to react selectively to certain stories but not others suggests a promising avenue for future research: exploring the kinds of news stories that compel people to reassess their preexisting beliefs. Consistent with this, recent studies have shown that when people are unfamiliar with specific issues, they are more likely to consume and share political information [32]. This suggests that the story about Biden's refugee policy in the current experiment likely caught respondents off guard, having a significant impact on both issue polarization and policy stances. It is also possible that most of the participants had already heard about election interference, lab leaks, and vaccines, and formed opinions, while the story about Cuban refugees was news to everyone.

Overall, these findings demonstrate that highlighting news stories covered from across the partisan divide can, in certain contexts, reduce polarization and the gaps in issue positions between Democrats and Republicans. Addressing "Blindspots" through this non-confrontational, information-sharing approach may be a promising method to mitigate polarization. Given that covering contentious political issues is unavoidable, this study suggests that those aiming to curb the rise of partisan polarization can do so by emphasizing particularly surprising stories that are overlooked by one political side. Such stories indicate that even for hot-button issues, the partisan divide is not insurmountable. Future studies aiming to reduce polarization could benefit from incorporating surprising political information.

## Supporting information

**S1 Appendix.**
(PDF)

## Author Contributions

**Conceptualization:** Curtis Bram.

**Investigation:** Curtis Bram.

**Writing – original draft:** Curtis Bram.

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
