## [Decision Letter · Decision Letter 0]

4 Sep 2023

PONE-D-23-17089Beyond Partisan Filters: Can Underreported News Reduce Issue Polarization?PLOS ONE

Dear Dr. Bram,

Thank you for submitting your manuscript to PLOS ONE. After careful consideration, we feel that it has merit but does not fully meet PLOS ONE’s publication criteria as it currently stands. Therefore, we invite you to submit a revised version of the manuscript that addresses the points raised during the review process.

We look forward to receiving your revised manuscript.

Kind regards,

Francesco Pierri, Ph.D.

Academic Editor

PLOS ONE

Journal Requirements:

7. Please ensure that you include a title page within your main document. You should list all authors and all affiliations as per our author instructions and clearly indicate the corresponding author.

8. Your ethics statement should only appear in the Methods section of your manuscript. If your ethics statement is written in any section besides the Methods, please move it to the Methods section and delete it from any other section. Please ensure that your ethics statement is included in your manuscript, as the ethics statement entered into the online submission form will not be published alongside your manuscript. 

9. We note that Figure 1 in your submission contain copyrighted images. All PLOS content is published under the Creative Commons Attribution License (CC BY 4.0), which means that the manuscript, images, and Supporting Information files will be freely available online, and any third party is permitted to access, download, copy, distribute, and use these materials in any way, even commercially, with proper attribution. For more information, see our copyright guidelines: http://journals.plos.org/plosone/s/licenses-and-copyright.

(1) You may seek permission from the original copyright holder of Figure 1 to publish the content specifically under the CC BY 4.0 license. 

(2) If you are unable to obtain permission from the original copyright holder to publish these figures under the CC BY 4.0 license or if the copyright holder’s requirements are incompatible with the CC BY 4.0 license, please either i) remove the figure or ii) supply a replacement figure that complies with the CC BY 4.0 license. Please check copyright information on all replacement figures and update the figure caption with source information. 

If applicable, please specify in the figure caption text when a figure is similar but not identical to the original image and is therefore for illustrative purposes only.

10. We note that Figure 1 & Supporting Figures S2 to S5 includes an image of a participant in the study. 

As per the PLOS ONE policy (http://journals.plos.org/plosone/s/submission-guidelines#loc-human-subjects-research) on papers that include identifying, or potentially identifying, information, the individual(s) or parent(s)/guardian(s) must be informed of the terms of the PLOS open-access (CC-BY) license and provide specific permission for publication of these details under the terms of this license. Please download the Consent Form for Publication in a PLOS Journal (http://journals.plos.org/plosone/s/file?id=8ce6/plos-consent-form-english.pdf). The signed consent form should not be submitted with the manuscript, but should be securely filed in the individual's case notes. 

Please amend the methods section and ethics statement of the manuscript to explicitly state that the participant has provided consent for publication: “The individual in this manuscript has given written informed consent (as outlined in PLOS consent form) to publish these case details”. 

11. Please include captions for your Supporting Information files at the end of your manuscript, and update any in-text citations to match accordingly. Please see our Supporting Information guidelines for more information: http://journals.plos.org/plosone/s/supporting-information. 

**Additional Editor Comments:**

Despite one negative review, I believe there is still room to improve the quality of the manuscript following reviewers' comments and meet the journal's standards of high quality.

Reviewers' comments:

Reviewer's Responses to Questions

**Comments to the Author**

1. Is the manuscript technically sound, and do the data support the conclusions?

Reviewer #1: Partly

Reviewer #2: Partly

2. Has the statistical analysis been performed appropriately and rigorously? 

Reviewer #1: Yes

Reviewer #2: Yes

3. Have the authors made all data underlying the findings in their manuscript fully available?

Reviewer #1: Yes

Reviewer #2: Yes

4. Is the manuscript presented in an intelligible fashion and written in standard English?

Reviewer #1: Yes

Reviewer #2: Yes

5. Review Comments to the Author

Reviewer #1: The study examines the effects of comprehensive information about coverage biases on mitigating the impact of partisan media on polarization. To do so, the author seeks attitudinal changes towards four breaking news stories typical of those overlooked by partisan media sources. More in detail, the experiment aimed to examine biases in understanding party positions using political stories (treatments) embedded within an emailed newsletter that respondents were neither compelled nor prompted to read. This newsletter clusters articles from diverse outlets into single stories to underscore coverage biases. Bias is rated at the source level using an average rating from 3 independent non-partisan media monitoring organizations.

The author analyzed four breaking stories from July 14th 2021, to July 18th 2021, that received limited coverage from conservative or liberal-leaning media. On July 19th, 2,372 subscribers to the Blindspot Report were invited to take a survey that included questions about all four issues, their ideology, country of origin, partisanship, and if they were from the US. 1861 subscribers reported an email address matching a Blindsport Report subscriber. Results show that only one of these stories may have mitigated polarization and influenced issue positions, underlying how exploring the conditions that determine the dynamics behind polarization from polarized media is crucial.

Despite the attractive experimental design, which included a newsletter analysis, the study lacks the quality that a PLOS paper deserves, leading me to reject it. From an overall perspective, the longitudinal analysis proposed analyzes only 4 days during 2021, which is a too-narrow analysis window when it comes to studying the news diet of users in relationship with their political stance. Furthermore, the introduction could have adequately covered many aspects described in the study (e.g., echo chambers and polarization on the online ecosystem), and the description of the experimental design lacks important technicalities that make the paper opinable nor reproducible. Below, you can find the major and minor changes I propose.

I hope these observations will encourage you to improve the work's quality and emphasize its potential better.

Major Changes

1. The introduction would benefit from a description of polarisation and how it relates to traditional and social media. I suggest looking at the works of Quattrociocchi, Etta, Cinelli, Lazer, De Domenico

2. The newsletter's algorithm needs more technicalities, and, at the same time, no previous studies cited used such a newsletter.

3. The lack of this information makes the experiment lacking soundness. Please, provide a better explanation of how this algorithm works and, if possible, include other studies that have already used this newsletter

4. Is bias rated on a discrete or continuous level? Please, provide a more detailed explanation of this rating

"Ground News' content suggests that its subscribers have an intrinsic interest in exploring narratives beyond conventional partisan lines." is this sentence supported by their statement? Please provide a link to that

5. For transparency purposes, the paper would benefit from a distribution of the biases of the newsletter on a broad timespan to quantitatively assess the objectiveness of the company itself

6. "Consequently, they could be more open to persuasion and exhibit more flexibility in their attitudes and beliefs than others." The lack of cited studies makes this sentence just an opinion from the author. I suggest bringing studies evidencing how the lack of a political leaning leads to better persuasion. Furthermore, the part "might exhibit more flexibility in their attitudes and beliefs than others." is too general. What does having a flexible attitude or belief mean? Furthermore, please provide studies that compare those with a "flexible attitude and belief" to those who don't have that

7. "I selected four breaking stories from July 14th 2021 to July 18th 2021 that received limited coverage from either conservative or liberal-leaning media." What does limited mean? Is there a coverage threshold or measure that defines a story as preferred from a specific political side from a more objective one?

8. "Of the initial 2,372 respondents, 1,861 reported an email address matching a Blindspot Report subscriber." I do not get the math here. If subscribers to the Blindspot Report were invited to take the survey, how can only 1861 of them be subscribers? They should have been 2372. Please, provide a better description of this.

9. Was the survey composed of open questions or closed ones? In the latter's case, what were the questions' interval ranges?

"In conclusion, these findings demonstrate that dismantling information echo chambers may, in certain contexts, reduce polarization and the gaps in issue positions between Democrats and Republicans." this sentence does not consider the concept of echo chambers. Here, the study does not account for the topology of the echo chambers. Therefore, no statement can be made about echo chambers but just the polarization itself.

Minor Changes

1. The motto of Fox News reads, Fair and Balanced," should have an opening quote instead of a comma

2. The New York Times proclaims, All the News That's Fit to Print." should have an opening quote instead of a comma. Moreover, the period at the end of the quote should be moved after the closing quote

3. "This raises the question: Can" should not have a capital C

4. "[…] either conservative- or liberal-leaning media." should not have a dash after conservative

5. "[…] were from the United States, their partisanship." should have the citing number before the period

6. The sentence "But ATEs are not the only useful way to analyze the data." should be removed since it looks too journalistic. The concept and motivations are already described in the following sentences. Just reframe them to make the reading smoother

7. Page 12," blindspot": please correct the opening quoting

Reviewer #2: This manuscript evaluates the results of a field study to assess if receiving information about issue bias may reduce political polarization and influence issue position. In the experiment, self-selected participants were exposed to coverage bias information on four underreported news topics in a newsletter, and their issue position and polarization stance were measured using a pre/post repeated measures survey design. I find this to be an important and compelling study, which in its current form has a some clarity issues and technical issues with how the hypotheses are set up and the results are discussed.

On the whole, I find this to be an interesting research question and a compelling study design. Many lab studies have shown the impacts of specific news interventions on issues like misinformation and polarization, but fewer are able to replicate them in the field. This is thus a valuable contribution as a field experiment, which the authors rightfully add to the list of field studies where the effect is harder to demonstrate than in a more controlled setting.

Additionally, I find that the focus on people who are already willing to have their minds changed is particularly interesting. As the authors state, this audience may be more malleable than the average population, which makes this a relevant population to target where we might expect significant effects.

The work is well-positioned within the extant literature. The authors may wish to acknowledge that the manipulation, being a one-time, small change may not have had a large impact on such important outcomes as issue polarization. For a related discussion, I find the discussion in Wojcieszak’s “Null effects of news exposure: a test of the (un)desirable effects of a ‘news vacation’ and ‘news binging’” to be a good discussion of how news constitute a small proportion of screen time and effects may thus be more challenging to measure in the field.

While I find this paper to be well presented and intelligible. there were a few areas I think should be clearer. As an improvement, I would suggest that the author make their hypotheses clearer — on page 7, they refer to “16 potential hypotheses under investigation,” but these are never explicitly outlined in the setup of the paper. The lack of clear hypotheses also make it challenging to interpret the second set of analyses presented from Page 12 onwards — are these additional, interesting exploratory analyses, or are they new contributions of this work to the literature? For additional clarity, the significant p-values used in table 1-4 should be aligned with the p-values discussed in the text (and p-values <0.1 should not be highlighted, in my opinion). Also the p-values thresholds used in table 5 should be stated in the table. There are also parts where the language feels too informal, such as a paragraph which begins with “But” on page 12 and uses of contractions (e.g. “doesn’t”). There are also missing quotation marks in the first sentence. While the experiment is well-described on page 6 and figure 1 aids comprehension, the setup is sufficiently complex that a reader would benefit from an additional figure or table to clarify which participants were exposed to which treatments.

I do not think the statistical analysis is set up in such a way as to make the findings most generalizable and interpretable. The author sets themselves up to assess 16 potential hypotheses using Bonferroni-adjusted p-values. The resulting threshold is rather stringent, and so the authors report results under both the Bonferroni-adjusted criteria and traditional 0.05 criteria. Overall, this makes the results challenging to interpret and detracts from the contribution of this paper. It is then unclear whether we should take this study as a) a study that continues a trend of null findings among field studies on news and their impact on attitudes, or b) a study with modest results that should be assessed and discussed as a contribution to the literature. Currently, the results section reads as a little bit of both, which makes it challenging to assess a clear contribution.

I believe one way to mitigate this would be to lead with an aggregated hypothesis and analysis that better sets up the broader question this paper is trying to answer. The broad question is whether exposure to information about the biased coverage of a news story can change perceptions of issue importance, issue position, issue polarization, and issue approval difference. Currently, the central hypotheses are split into 16 parts: 4 per issue x 4 per outcome, which significantly reduces the Bonferroni-adjusted p-value under which the author would accept a significant result. It is not clear to me why the author would lead with the issue-specific findings when they are looking for an overall trend. I might instead consider foregrounding an analysis more similar to that from Table 5 as the main contribution, which has the outcome variable aggregated across all four issues. This would reduce the number of hypotheses that need to be tested and adjusted for. The specific story breakdowns would then serve more as an exploratory analysis of how this effect might look different for different issues. Again, I think a clear presentation of the hypotheses would also make the contributions of the analyses easier to assess.

6. PLOS authors have the option to publish the peer review history of their article (what does this mean?). If published, this will include your full peer review and any attached files.

Reviewer #1: No

Reviewer #2: No

---

## [Author Response · Author response to Decision Letter 0]

20 Oct 2023

Please see the included document for the properly formatted version of my response to reviewers and editors.

---

## [Decision Letter · Decision Letter 1]

11 Dec 2023

PONE-D-23-17089R1Beyond Partisan Filters: Can Underreported News Reduce Issue Polarization?PLOS ONE

Dear Dr. Bram,

Thank you for submitting your manuscript to PLOS ONE. After careful consideration, we feel that it has merit but does not fully meet PLOS ONE’s publication criteria as it currently stands. Both reviewers are positive about the revision, but still needs some revisions. Therefore, we invite you to submit a revised version of the manuscript that addresses the points raised during the review process. 

We look forward to receiving your revised manuscript.

Kind regards,

Yongjun Zhang

Academic Editor

PLOS ONE

Journal Requirements:

Reviewers' comments:

Reviewer's Responses to Questions

**Comments to the Author**

1. If the authors have adequately addressed your comments raised in a previous round of review and you feel that this manuscript is now acceptable for publication, you may indicate that here to bypass the “Comments to the Author” section, enter your conflict of interest statement in the “Confidential to Editor” section, and submit your "Accept" recommendation.

Reviewer #2: (No Response)

Reviewer #3: (No Response)

2. Is the manuscript technically sound, and do the data support the conclusions?

Reviewer #2: Yes

Reviewer #3: Yes

3. Has the statistical analysis been performed appropriately and rigorously? 

Reviewer #2: Yes

Reviewer #3: Yes

4. Have the authors made all data underlying the findings in their manuscript fully available?

Reviewer #2: Yes

Reviewer #3: Yes

5. Is the manuscript presented in an intelligible fashion and written in standard English?

Reviewer #2: Yes

Reviewer #3: Yes

6. Review Comments to the Author

Reviewer #2: I thank the author for incorporating the comments from reviewers. I find the manuscript much improved, with a few remaining areas for improvement. Overall, I think the paper and findings are both clearer and cleaner, which is fantastic. The new framing around the results make this a neat and targeted jumping-off point for future research, and I admire the author having run a field experiment at scale. I also thought the discussion was significantly strengthened and better helped to understand how to interpret these findings in context.

The analysis is much clearer and I think the de-emphasis of the ATE analyses serves to hone the central message of the paper. My one remaining concern, which I initially missed in the first revision, is that the R^2 and adjusted R^2 appear to be near-zero for all models in Table 1. While the R^2 is of course not the only way to assess goodness-of-fit, and does not impact the coefficient effect significance, such a low R^2 value is quite concerning to me. I am wondering if the author could explain why they believe the R^2 value to be so low? Perhaps they could try to account for demographic variables as an additional control? This can be added to the appendix if they do not lead to a substantially different conclusion.

While overall I like the changes in the introduction and discussion, recent studies about echo chambers/filter bubbles have found mixed evidence that people truly live in online echo chambers. (e.g. “Echo chambers, filter bubbles, and polarisation: a literature review”, by Arguedas et al. 2022, or Dahlgren et al., 2021, for a comparison between filter bubbles and selective exposure theories, “A critical review of filter bubbles and a comparison with selective exposure.”) Though I think it is fine to motivate the work through the echo chamber lens, I think more hedging/critical language when talking about echo chambers, or just a nod to the fact that these theories are contested would serve the paper well. Alternatively (or in parallel), strengthening the selective partisan exposure piece of the motivation (lines 31-36) would also address these concerns.

A small, personal nitpick is that in the paragraph lines 195-208, the irreferential word “This” is used five times to start a new sentence. This practice generally makes arguments harder to follow (e.g. the difference between starting the current sentence with “This makes arguments hard to follow” vs “This practice makes arguments harder to follow.”)

Reviewer #3: I carefully read the resubmitted version and the response memo to editors and reviewers. The revised manuscript provides a clear statement on motivation, research design, and interpretation on the results. I believe that the authors handled most of the reviewers’ comments well. Since I did not review the original manucript, I provide two additional comments based on the resubmitted version.

First, I think the authors should be more cautious about the selection of news articles serving as instruments. Reviewer 1 has provided some comments on algrothm of the newsletter and selection of news articles. I agree with these comments. After I looked at the letter, I doubt that the author responded these comments well. The author should provide details on how the newsletter rated the news stories using “natural language process”. This is crucial to the paper because the ideological spectrum is the basis of designing experiment instruments. Also, the author argued that “the selection of stories was ultimately based on my judge,emt about issue salience at the time, a point which I am now clearer about in the paper”. I would argue that this selection criteria could also affect the experimental results because it was likely that different set of news stories could lead to different results (possibly). I would suggest that a more clear and detailed statements on the limitation of the selection of news articles would be necessary.

Second, some statements in the article are not convincing. For instance, the author stated “one can imagine much larger effects were people to consistently receive information about their Blindspots on one or more issues over time”. This statememt should be based on previous research which concluded the larger effect of repeated exposure to biased/selected information on attitudes. Otherwise, this statement is not convincing.

7. PLOS authors have the option to publish the peer review history of their article (what does this mean?). If published, this will include your full peer review and any attached files.

Reviewer #2: No

Reviewer #3: No

---

## [Author Response · Author response to Decision Letter 1]

8 Jan 2024

Dear Editors and Reviewers,

Thank you for the opportunity to revise and resubmit my manuscript and for the detailed comments. 

Turning directly to the comments from Reviewer #2, I completely agree that the low R^2 is a concern when interpreting these results. As suggested by this reviewer, the Appendix now includes an additional section which incorporates demographic controls into the main analysis. I do not find that this makes a meaningful difference in the share of variation explained. I now flag the low R^2 numbers in the main text to ensure readers are aware of this issue.

I also agree that my discussion of echo chambers and selective exposure did not incorporate important nuance in research on these topics. The citations suggested by this reviewer were helpful and I now incorporate both in the introduction. Finally, I have corrected the overuse of the word “This” in the paragraph mentioned. 

Turning to comments from Reviewer #3. I agree that using my judgement to select the stories may have affected the results, and that a more systematic way of selecting stories would be a natural next step for this experimental design. I have now clearly noted this in the manuscript so that readers are aware of this important limitation. Ground News’s algorithms are proprietary, a point which I also now note in the manuscript. 

Finally, I have gone through the manuscript again and have attempted to soften unconvincing language suggested by this reviewer. I revised the speculation about effects in a hypothetical longer term study to say that effects may be different, not necessarily “much larger” as I had before. 

Once again, thank you all for your time and attention and I am happy to answer any questions. I look forward to hearing from you.

Sincerely,

Curtis Bram

Assistant Professor 

UT Dallas

---

## [Editor Report · Decision Letter 2]

15 Jan 2024

Beyond Partisan Filters: Can Underreported News Reduce Issue Polarization?

PONE-D-23-17089R2

Dear Dr. Bram,

We’re pleased to inform you that your manuscript has been judged scientifically suitable for publication and will be formally accepted for publication once it meets all outstanding technical requirements.

Kind regards,

Yongjun Zhang

Academic Editor

PLOS ONE

Additional Editor Comments (optional):

After reviewing your paper and response memo, I believe this paper meets PLOS One's publication criteria and can make a great contribution to the current literature. 
---

## [Editor Report · Acceptance letter]

7 Feb 2024

PONE-D-23-17089R2 

PLOS ONE

Dear Dr. Bram, 

I'm pleased to inform you that your manuscript has been deemed suitable for publication in PLOS ONE. Congratulations! Your manuscript is now being handed over to our production team.

Kind regards, 

on behalf of

Dr. Yongjun Zhang 

Academic Editor

PLOS ONE